# Auxin regulates SNARE-dependent vacuolar morphology restricting cell size

Christian Löfke, Kai Dünser, David Scheuring, Jürgen Kleine-Vehn*

Department of Applied Genetics and Cell Biology, University of Natural Resources and Life Sciences, Vienna, Austria

**Abstract** The control of cellular growth is central to multicellular patterning. In plants, the encapsulating cell wall literally binds neighbouring cells to each other and limits cellular sliding/migration. In contrast to its developmental importance, growth regulation is poorly understood in plants. Here, we reveal that the phytohormone auxin impacts on the shape of the biggest plant organelle, the vacuole. TIR1/AFBs-dependent auxin signalling posttranslationally controls the protein abundance of vacuolar SNARE components. Genetic and pharmacological interference with the auxin effect on vacuolar SNAREs interrelates with auxin-resistant vacuolar morphogenesis and cell size regulation. Vacuolar SNARE VTI11 is strictly required for auxin-reliant vacuolar morphogenesis and loss of function renders cells largely insensitive to auxin-dependent growth inhibition. Our data suggests that the adaptation of SNARE-dependent vacuolar morphogenesis allows auxin to limit cellular expansion, contributing to root organ growth rates.

## Introduction

Symplastic growth, characterised by cells that do not alter their relative position to each other, is typical in plant tissue expansion (*Priestley, 1930*; *Erickson, 1986*). Such development implies supra-cellular (above the level of single cells) regulation, which has an enormous impact on cellular growth control for plant patterning. Despite their importance, molecular mechanisms that restrict cellular and tissue growth are poorly understood in plants.

The phytohormone auxin is a crucial growth regulator and central in differential growth processes. TRANSPORT INHIBITOR RESISTANT1 (TIR1) and its homologs AUXIN F-BOX PROTEINS (AFBs) have been unequivocally demonstrated to be auxin receptors (*Kepinski and Leyser, 2005*; *Dharmasiri et al., 2005a*). Genomic auxin responses are initiated by auxin binding to TIR1/AFBs, promoting its interaction with proteins of the AUXIN/INDOLE ACETIC ACID (Aux/IAA) family. Auxin-dependent formation of such a co-receptor pair triggers the ubiquitination and subsequent degradation of Aux/IAA proteins. In the absence of auxin, Aux/IAAs form inhibitory heterodimers with AUXIN RESPONSE FACTOR (ARF) family transcription factors. Thus, auxin-dependent Aux/IAA degradation leads to the release of ARF transcription factors and subsequent transcriptional responses (for reviews, see *Quint and Gray, 2006*; *Sauer et al., 2013*).

Intriguingly, auxin-signalling events promote and inhibit cellular growth in a cell-type- and auxin concentration-dependent manner. Physiological auxin levels induce growth in light grown aerial tissues, while most root tissues show growth repression in response to the very same auxin concentrations (*Sauer et al., 2013*). The 'acid growth' theory proposes that auxin causes extracellular acidifications and subsequent cell wall remodelling, ultimately driving turgor-dependent cellular expansion (*Sauer and Kleine-Vehn, 2011*). This theory is based on tissues showing auxin-dependent growth induction. In contrast, relatively little is known of how auxin inhibits cellular growth in other tissues.

The higher plant vacuole is, due to its size, the most prominent plant organelle, and shares its lytic function with its counterparts in yeast and animal lysosomes (*Marty, 1999*). It may likewise be

*For correspondence: juergen.kleine-vehn@boku.ac.at

Competing interests: The authors declare that no competing interests exist.

**eLife digest** In plants and animals, the way that cells grow is carefully controlled to enable tissues and organs to form and be maintained. This is especially important in plants because the cells are attached to each other by their cell walls. This means that, unlike some animal cells, plant cells are not able to move around as a plant's organs develop.

Plant cells contain a large storage compartment called the vacuole, which occupies 30–80% of a cell's volume. The volume of the vacuole increases as the cell increases in size, and some researchers have suggested that the vacuole might help to control cell growth. A plant hormone called auxin can alter the growth of plant cells. However, this hormone's effect depends on the position of the cell in the plant; for example, it inhibits the growth of root cells, but promotes the growth of cells in the shoots and leaves. Nevertheless, it is not clear precisely how auxin controls plant cell growth.

Here, Löfke et al. studied the effect of auxin on the appearance of vacuoles in a type of plant cell—called the root epidermal cell—on the surface of the roots of a plant called *Arabidopsis thaliana*. The experiments show that auxin alters the appearance of the vacuoles in these cells so they become smaller in size. At the same time, auxin also inhibits the growth of these cells.

Löfke et al. found that auxin increases the amount of certain proteins in the membrane that surrounds the vacuole. These proteins belong to the SNARE family and one SNARE protein in particular, called VTI11, is required for auxin to be able to both alter the appearance of the vacuoles and restrict the growth of root epidermal cells. Enzymes called PI4 kinases were also shown to be involved in the control of the SNARE proteins in response to the auxin hormone.

Löfke et al.'s findings suggest that auxin restricts the growth of root epidermal cells by controlling the amount of SNARE proteins in the vacuole membrane. The next challenge will be to understand precisely how the shape of the vacuole is controlled and how it contributes to cell growth.

hypothesised that multifunctional plant vacuoles also contribute to cellular size regulation, as the volume of vacuoles correlates with individual cell size in plant cell cultures (*Owens and Poole, 1979*). The root epidermis is a suitable model to study such processes (*Löfke et al., 2013*), because it is regularly spaced into shorter tricho- and longer atrichoblast cell files in the late meristematic zone (*Berger et al., 1998*), which intriguingly, show smaller and larger vacuolar structures, respectively (*Berger et al., 1998*; *Löfke et al., 2013*). However, the functional importance of this correlation remains to be addressed.

Here we use this cell biological model system to reveal and subsequently characterise auxin-dependent vacuolar morphogenesis and its requirement for limiting cellular growth.

## Results

### Auxin impacts on vacuolar morphology

Epidermal cells show smaller and larger vacuolar structures in shorter tricho- and longer atrichoblast cells, respectively (*Berger et al., 1998*; *Löfke et al., 2013*) (*Figure 1A*). We hypothesised that if the vacuolar morphology contributes to cellular size, the growth regulator auxin may impact on its regulation.

To specifically investigate the role of auxin in limiting cellular size in roots, we initially assessed how auxin impacts on late meristematic epidermal cells. To allow cellular development under high auxin conditions, we exogenously applied synthetic auxin, 1-naphtylacetic acid (NAA), in nanomolar ranges for 20 hr and screened for subcellular effects that showed differential regulation in neighbouring cells. Remarkably, exogenous application of auxin (NAA [250 nM]) or endogenous elevation of YUCCA-dependent auxin biosynthesis led to a dramatic change in vacuolar appearance in root epidermal cells, which consequently displayed smaller luminal vacuolar structures (*Figure 1A,B,E,F*).

We established a vacuolar morphology index in epidermal cells based on the biggest luminal structure to further evaluate the apparent auxin effect on vacuolar shape (*Figure 1—figure supplement 1*). This analysis revealed that high auxin conditions affect vacuolar structures, particularly in atrichoblasts (*Figure 1D,G*). Adversely, pharmacological depletion of auxin caused visibly larger

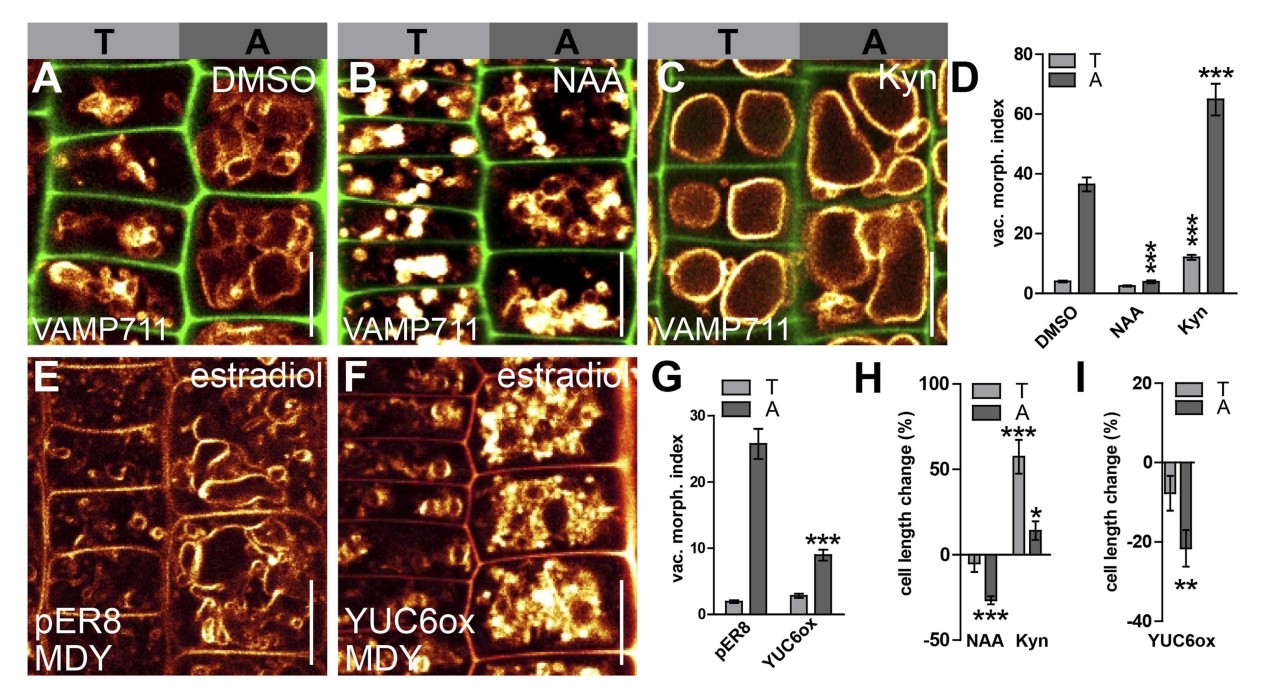

**Figure 1**. Auxin triggered changes in vacuolar morphology correlate with its effect on cell size. (**A–C**) Seedlings treated with the solvent DMSO (**A**), auxin (**B**) (NAA 250 nM; 20 hr) or auxin biosynthesis inhibitor kynurenin (**C**) (Kyn) (2 µM; 20 hr). Tonoplast localised VAMP711-YFP (orange) as vacuolar marker and propidium iodide stain (green) for decorating the cell wall were used for confocal imaging of tricho-/atrichoblast (T/A) cells (**A–C**). (**D**) Vacuolar morphology (vac. morph. [µm²]) index of tricho/atrichoblast cells after pharmacological manipulation of auxin levels. (**E–G**) Vacuolar morphology of estradiol (10 µM; 20 hr) induced YUCCA6 overexpression (YUC6ox) (**F**) and the respective empty vector control (pER8) (**E**). Tonoplast membrane stain MDY-64 (orange) was used for confocal imaging. (**G**) Vacuolar morphology (vac. morph. [µm²]) index after genetic manipulation of auxin levels. (**H** and **I**) Quantification of cell length change in tricho-/atrichoblast (T/A) cells following pharmacological (**H**) or genetic manipulation of auxin levels (**I**). For statistical analysis, treated cells were compared to untreated tricho-/atrichoblast. n = 40 cells in 10 individual seedlings for cell length measurement and n = 40 cells in eight individual seedlings for vacuolar morphology index quantification. Error bars represent s.e.m. Student's t-test p-values: *p < 0.05, **p < 0.01, ***p < 0.001. Scale bar: 15 µm (**A–C**, **F**, **G**).

The following figure supplements are available for figure 1:

**Figure supplement 1**. Quantification of epidermal cell length and vacuolar morphology.

**Figure supplement 2**. Auxin does not affect vacuolar morphology of epidermal root cells in the differentiation zone.

vacuolar structures in both cell types, but was more pronounced in trichoblasts (relative to the untreated control) (*Figure 1A,C,D*).

This data shows that auxin differentially affects vacuolar shape in neighbouring epidermal cells.

Notably, the differential effect of auxin on vacuoles correlated exactly with a differential effect on cellular size. High auxin levels reduced the cell length of atrichoblast cells (*Figure 1A,B,E,F,H,I*), whereas cell lengths of the smaller trichoblasts were not significantly affected (*Figure 1A,B,E,F,H,I*). Conversely, pharmacological reduction in auxin biosynthesis mainly increased the cell length in trichoblasts (*Figure 1A,C,H*).

Our data shows that the auxin effect on vacuolar morphology correlates with its negative effect on late meristematic epidermal cell size. Auxin treatments manifestly did not reverse vacuolar morphology of fully elongated epidermal root cells in the differentiation zone (*Figure 1—figure supplement 2*). This finding suggests that auxin mainly shapes vacuoles in growth competent cells.

Next we investigated the auxin effect on vacuoles in the course of time. As the auxin effect on vacuoles was most pronounced in atrichoblasts, we focused our analysis (from here onwards) mainly on this cell-type. Notably, auxin imposed in time steadily increasing effects on vacuolar appearance (*Figure 2—figure supplement 1*). Auxin induced detectable changes in vacuolar morphology already

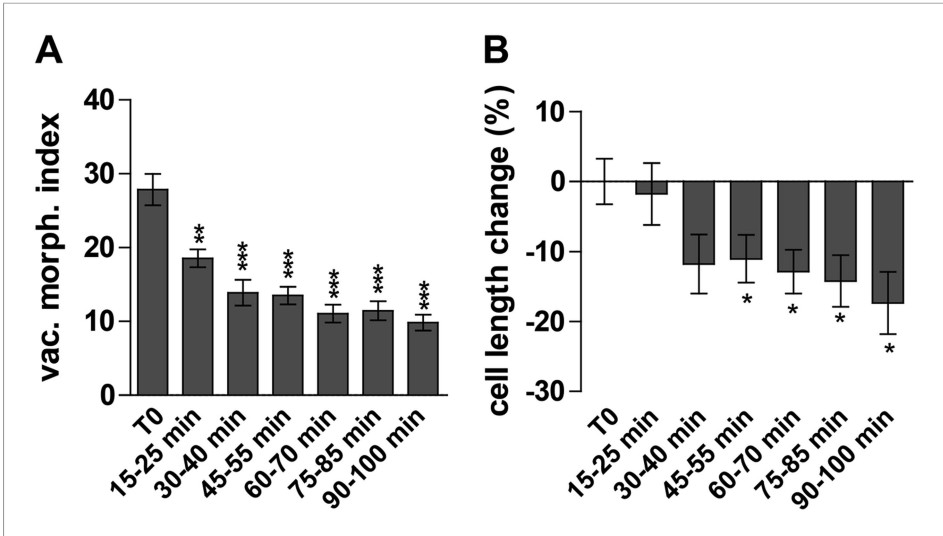

**Figure 2**. Auxin effect on vacuoles precedes cell size regulation. (**A** and **B**) Time course imaging of 250 nM NAA treated seedlings were performed every 15 min. Image acquisition took 10 min per time point. Graphs depict vacuolar morphology index (**A**) and cell length of atrichoblasts (**B**). Untreated seedlings were imaged before and after recording the auxin treated samples and resulting average was defined as T0. Error bars represent s.e.m. For statistical analysis DMSO and NAA treatments were compared. n = 50 cells in 10 individual seedlings for each time point. Student's *t*-test p-values: *p < 0.05 **p < 0.01, ***p < 0.001.

The following figure supplement is available for figure 2:

**Figure supplement 1**. Auxin effect on vacuolar morphology increases in time.

after 15–30 min (*Figure 2A*). On the other hand the auxin effect on late meristematic cell size was slightly later starting to be significantly affected around 45 min (*Figure 2B*).

Based on our time course experiments we conclude that the auxin effect on vacuolar morphology precedes the auxin impact on late meristematic cell size.

## TIR1/AFBs-dependent auxin signalling triggers alterations in vacuolar morphology

We subsequently further characterised the unprecedented role of auxin in vacuolar morphogenesis. The auxin effect on vacuoles was in the time frame of fast transcriptional responses and, subsequently, we tested whether the TRANSPORT INHIBITOR RESISTANT1 (TIR1)/AUXIN F-BOX PROTEINS (AFB) auxin receptors (*Leyser, 2006*; *Mockaitis and Estelle, 2008*) are required for the auxin effect on vacuoles. It has been suggested that auxin analogue 5-F-IAA preferentially triggers genomic auxin responses via TIR1/AFBs (*Robert et al., 2010*) and it indeed induced small luminal vacuoles (*Figure 3A,B,E*). Correspondingly, auxinole, a designated inhibitor of TIR1/AFBs auxin receptors (*Hayashi et al., 2012*), blocked the auxin effect on vacuolar morphology (*Figure 3A,C,D,E*), and, the genetic reduction of TIR1/AFBs functions in *tir1 afb1 afb3* triple mutants prompted partial resistance to the auxin-induced changes in vacuolar appearance (*Figure 3F–J*).

This set of data indicates that TIR1/AFBs-dependent auxin signalling is required for the auxin effect on vacuolar morphogenesis.

## TIR1/AFBs-dependent auxin perception posttranslationally stabilises vacuolar SNAREs

In the following we got interested in **SNA**P (Soluble NSF Attachment Protein) **Re**ceptor (SNARE) complexes at the vacuole. Proximity of adjacent membrane allows the interaction of v (vesicle)- and t (target)-SNAREs to form a complex, allowing the fusion of vesicles to specific target membranes. SNAREs are essential for eukaryotic vesicle trafficking and according to structural features SNAREs are

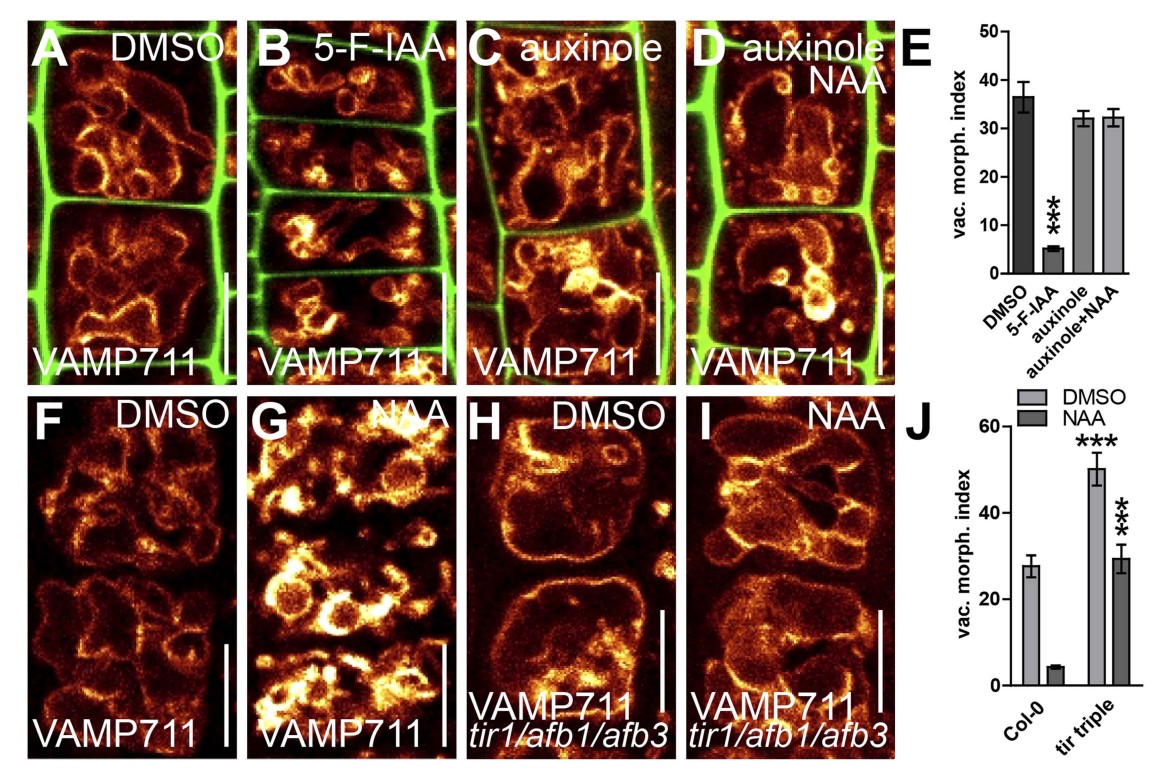

**Figure 3**. Auxin affects vacuolar morphology in a TIR1/AFBs-dependent manner. (**A–D**) Seedlings treated with DMSO (**A**), auxin analogue 5-F-IAA (**B**) (250 nM; 20 hr), TIR1/AFBs antagonist auxinole (**C**) (20 μM; 20 hr) and concomitant with NAA and auxinole (**D**). Tonoplast localised VAMP711-YFP (orange) as vacuolar marker and propidium iodide (green) for decorating the cell wall was used for confocal imaging (**A–D**). (**E**) Vacuolar morphology (vac. morph. [μm²]) index of treatments used in **A–D**. For statistical analysis DMSO and treatments were compared. (**F–I**) DMSO (**F**) or NAA (**G**) (250 nM; 20 hr) treated control seedlings compared to *tir/afb1/afb3 triple* mutants treated with DMSO (**H**) or NAA (**I**) (250 nM; 20 hr). Tonoplast localised VAMP711-RFP (orange) as vacuolar marker was used for confocal imaging in **F–I**. (**J**) Vacuolar morphology (vac. morph. [μm²]) index of treatments shown in **F–I**. For statistical analysis either DMSO or NAA treatments were compared between control and indicated mutant. n= 40 cells in eight individual seedlings. Error bars represent s.e.m. Student's *t*-test P-values: ***$p < 0.001$. Scale bar: 15 μm.

divided in R (arginine)- and Q (glutamine)-SNAREs (*Martens and McMahon, 2008*). In yeast, the SNARE complex is furthermore central in homotypic vacuolar membrane remodelling and proteomic approaches have identified conserved SNARE complexes at the plant tonoplast (*Carter et al., 2004*). Ergo, we tested whether auxin affects vacuolar SNAREs in *Arabidopsis*. Remarkably, increased auxin biosynthesis or exogenous application of auxin increased the fluorescence intensity of tonoplast localised SNAREs, such as VAMP711-YFP, SYP21-YFP and SYP22-GFP (*Figure 4A–L,N,O*). Auxin severely impacts on vacuolar appearance and, hence, it could be that membrane crowding induces higher fluorescence. To address this question we performed co-localisation of VAMP711-RFP/YFP and membrane dyes, such as FM4-64 and MDY-64. Notably, VAMP711-RFP/YFP, but not the membrane dyes showed auxin-induced signal intensities, suggesting that the auxin effect on vacuolar SNAREs does not rely on membrane crowding (*Figure 4—figure supplement 1*).

Exogenous application of auxin does not detectably impact on cytosolic pH (*Gjetting et al., 2012*). However, to fully preclude that pH sensitivity may affect fluorescence of VAMP711-YFP (YFP faces the cytosol), we also utilized more pH resistant RFP fusions. The auxin effect on VAMP711 was detectable in both pH sensitive YFP and pH resistant RFP fusions (*Figure 4—figure supplement 1A,B*; *Figure 5A,B*), suggesting that the auxin effect on SNAREs does not indirectly rely on cytosolic pH.

To assess whether auxin treatments affect the overall cellular abundance of vacuolar SNAREs, we performed defined z-stack imaging. Subsequent maximum projections and intensity measurements

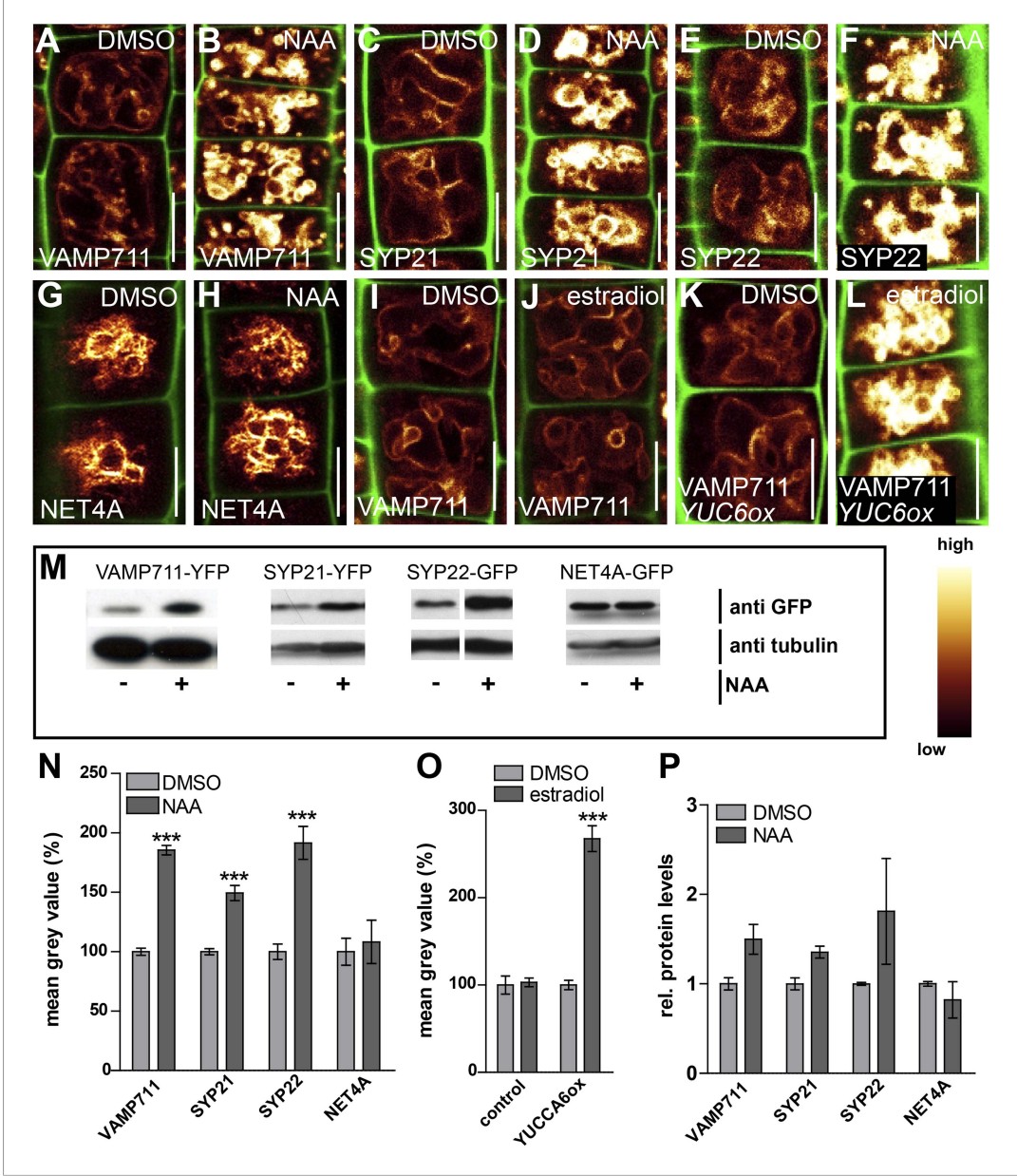

**Figure 4**. Auxin posttranslationally stabilises tonoplast localised SNAREs. (**A–L**) Tonoplast localised SNAREs and marker protein NET4A under high auxin conditions. *pUBQ10::VAMP711-YFP* (**A** and **B**), *35S::SYP21-YFP* (**C** and **D**), *SYP22::SYP22-GFP* (in *syp22*) (**E** and **F**) and *NET4A::NET4A-GFP* (**G** and **H**) expressing seedlings treated with DMSO (**A**, **C**, **E**, **G**) or NAA (**B**, **D**, **F**, **H**) (500 nM; 20 hr). (**I** and **J**) Estradiol does not affect VAMP711-YFP abundance. *pUBQ10::VAMP711-YFP* expressing seedlings treated with DMSO (**I**) or estradiol (**J**) (10 μM estradiol; 20 hr). YUCCA6 expression under control of an estradiol inducible promoter in *pUBQ10::VAMP711-YFP* expressing seedlings after DMSO (**K**) or estradiol (**L**) (10 μM; 20 hr) treatment. Propidium iodide (green) for decorating the cell wall was used for confocal imaging of atrichoblast cells. (**M**) Western-blot (anti-GFP) representing VAMP711-YFP, SYP21-YFP, SYP22-GFP and NET4A-GFP protein abundance after NAA (500 nM; 20 hr) and control treatment as well as corresponding alpha-tubulin abundance for normalization. (**N**) Mean grey value of vacuolar localised SNAREs and marker protein NET4A after auxin treatments (500 nM; 20 hr) compared to DMSO treatments. (**O**) Mean grey value of VAMP711-YFP after YUCCA6 induction (10 μM; 20 hr). (**P**) Western-blot quantification (mean grey values). n = 3 biological replicates each consisting of a pool of 40–50 roots. Error bars represent s.e.m. For statistical analysis DMSO and NAA

*Figure 4. Continued*
treatments were compared. For confocal analysis (**N**-**O**): n = 32 cells in eight individual seedlings. Student's *t*-test p-values: ***p < 0.001. Scale bar: 15 μm.
The following figure supplements are available for figure 4:

**Figure supplement 1**. Increase in SNARE intensity is independent of membrane crowding.

**Figure supplement 2**. Auxin affects cellular abundance of vacuolar SNAREs.

**Figure supplement 3**. Induction of a single SNARE component is not sufficient to affect vacuolar morphology.

confirmed that auxin increases cellular SNARE abundance at the tonoplast (*Figure 4—figure supplement 2A–I*). To further emphasize on this finding, we also performed western blots on excised root tips, similarly confirming our conclusion that auxin increases vacuolar SNARE abundance (*Figure 4M,P*).

Notably, tonoplast marker NET4A-GFP (*Deeks et al., 2012*) did not show auxin-induced stabilization (*Figure 4G,H*; *Figure 4—figure supplement 2G,H*), suggesting certain specificity for the auxin effect on vacuolar SNAREs. This set of data indicates that auxin affects vacuolar SNARE function.

Induction of a single SNARE component, such as VAMP711, did not affect vacuolar morphogenesis (*Figure 4—figure supplement 3*), possibly indicating the joint requirement of several complex components. Furthermore, auxin modulated SNARE proteins also when expressed under constitutive promoters (*Figure 4A–D,I–N*). This data implies that auxin affects the vacuolar SNAREs posttranslationally.

Next we tested whether TIR1/AFBs-dependent auxin perception mechanisms are required for the auxin effect on vacuolar SNAREs. Concomitant treatments with auxin and the TIR1/AFBs antagonist auxinole comprehensively interfered with auxin-induced stabilisation of VAMP711-YFP (*Figure 5A–E*). Concurrently, the auxin effect on vacuolar SNAREs was also significantly reduced in the *tir1 afb1 afb3* triple mutant (*Figure 5F–J*). Hence, pharmacologic and genetic interference with TIR1/AFBs did not only inhibit the auxin effect on vacuoles, but also abolished the posttranslational effect of auxin on VAMP711.

We conclude that the TIR1/AFBs-dependent auxin signalling triggers higher SNARE abundance at the tonoplast.

## Vacuolar SNARE VTI11 function is required for the auxin-dependent modulation of vacuolar morphology

It has been suggested that several vacuolar SNARE components act redundantly (*Yano et al., 2003*; *Uemura et al., 2010*) and also in our conditions most analysed *SNARE* single mutants displayed vacuolar morphology reminiscent to wild type (*Figure 6—figure supplement 1*). In contrast, *vti11* mutant alleles display roundish vacuoles in untreated conditions (*Yano et al., 2003*; *Zheng et al., 2014*) (*Figure 6A,C*). Despite these apparent defects, vacuoles remained differentially controlled in *vti11* mutant tricho- and atrichoblast cells (*Figure 6—figure supplement 2*), indicating that the cell type-dependent regulation of vacuolar morphology is at least partially operational in *vti11* mutants.

We, hence, have chosen *vti11* mutants for further investigation and tested if VTI11 function is required for the auxin effect on vacuoles. Auxin treatments were less effective to modulate vacuolar morphology in *vti11* mutants (*Figure 6A–E*). Notably, pVTI11:VTI11-GFP expression in *vti11* mutant cells induced reversion to auxin sensitive vacuolar morphology (*Figure 6—figure supplement 3*). This data indicates that auxin does not only affect SNARE abundance, but requires functional Q-SNARE VTI11 to modulate vacuolar shapes.

Vacuoles partially escaped auxin regulation in *vti11* mutants, allowing us to assess the requirement of VTI11 function for auxin-dependent limitation of meristematic cell size. Interestingly, *vti11* mutants were not only partially resistant to the auxin effect on vacuoles, but in addition, less sensitive to the negative impact of auxin on late meristematic cell size (*Figure 6A–D,F*). This data suggests that VTI11 function is required for the auxin effect on vacuolar shape and late meristematic cell size.

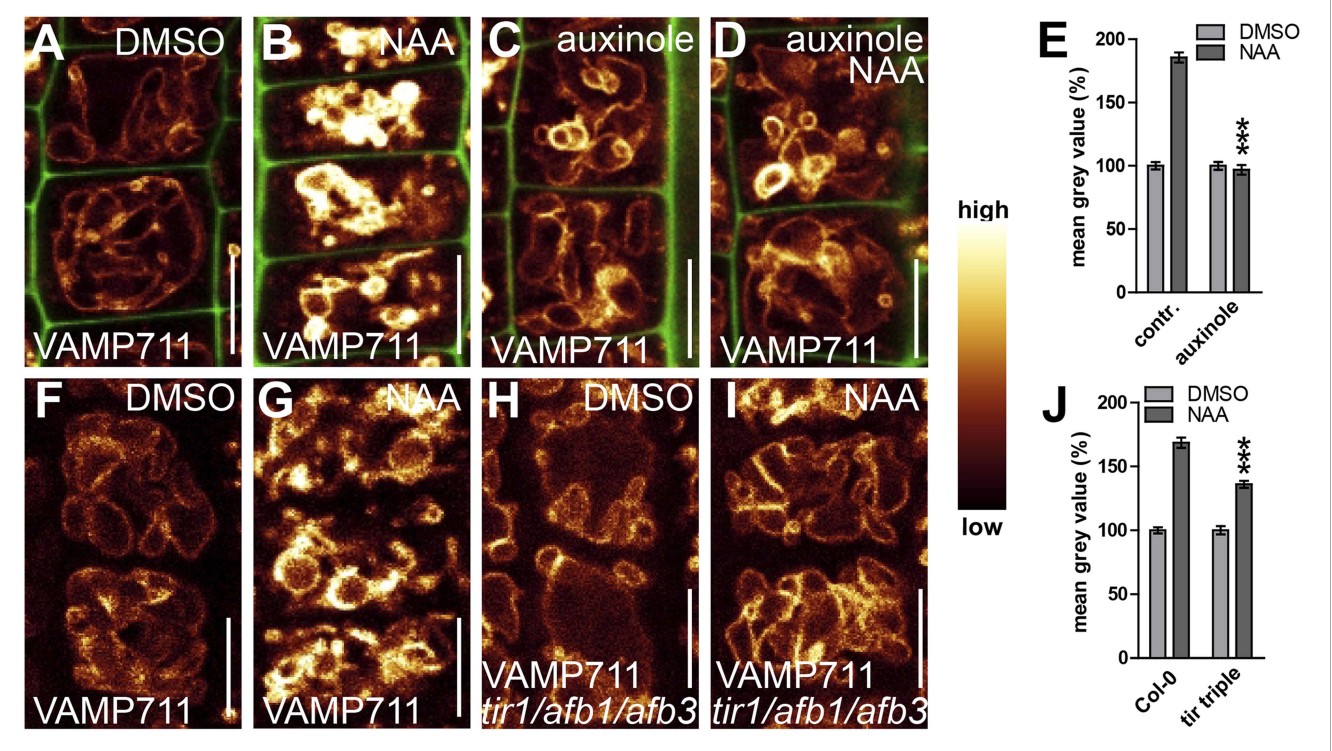

**Figure 5**. TIR1/AFBs-dependent auxin perception is required for posttranslational VAMP711 stabilisation. (**A–D**) Pharmacological inhibition of TIR1/AFBs dependent signalling. *pUBQ10::VAMP711-YFP* expressing seedlings treated with DMSO (**A**), NAA (**B**) (500 nM; 20 hr), auxinole (**C**) (20 µM; 20 hr) or auxinole/NAA co-treatment (**D**). (**E**) Mean grey value of VAMP711-YFP abundance after NAA or NAA/auxinole co-treatments. (**F–I**) Genetic inhibition of TIR1/AFBs signalling. VAMP711-RFP expressing control seedlings (for **H** and **I** treated with DMSO (**F**) and NAA (**G**) (500 nM; 20 hr). VAMP711-RFP abundance in *tir1-1/afb1-3/afb3-4* mutant background after DMSO (**H**) or NAA (**I**) (500 nM; 20 hr) treatment. (**J**) Mean grey value of treatments in **F–I**. VAMP711-YFP/RFP (orange) as a vacuolar marker and propidium iodide (green) for decorating the cell wall were used for confocal imaging of atrichoblast cells. n = 32 cells in eight individual seedlings. Error bars represent s.e.m. For statistical analysis either DMSO or NAA treatments were compared between control and indicated mutant/treated seedlings. Student's *t*-test p-values: ***p < 0.001. Scale bar: 15 µm.

## Interference with phosphatidylinositol homeostasis affects vacuolar SNAREs and impedes auxin-dependent cell size control

Several phosphatidylinositol (PI) -dependent processes have been previously shown to play a role in vacuolar biogenesis in yeast (*Mayer et al., 2000*) and also impact on vacuolar morphology in plants (*Nováková et al., 2014*; *Zheng et al., 2014*). PI3/4 kinase inhibitor Wortmannin (WM) affects vacuolar morphology and has been recently proffered as affecting processes upstream of vacuolar SNAREs in plants (*Feraru et al., 2010*; *Zheng et al., 2014*). WM treatments led to larger luminal vacuoles and abolished the auxin effect on vacuoles (*Figure 7A–E*). It may be noted that the negative effect of auxin on limiting late meristematic cell size was also abolished after low doses of WM [2 µM] (*Figure 7F*). This data suggests that WM sensitive processes may contribute to auxin-dependent vacuolar morphogenesis and cell size regulation.

To substantiate this pharmacological data, we subsequently screened the relevant literature for WM sensitive molecular components, which may affect root epidermal processes. PI4Kß1 and PI4Kß2 are expressed in root epidermal cells and redundantly control mature root hair morphology (*Preuss et al., 2006*). We therefore tested the auxin effect on vacuolar morphology in *pi4kß1/2* double mutants. Compared to wild-type seedlings, *pi4kß1/2* double mutants showed partially auxin resistant vacuolar appearance (*Figure 7H–L*). Moreover, the genetic interference with PI4Kß-function was reminiscent of WM treatments and also abolished the auxin effect on reducing late meristematic cell size (*Figure 7M*).

Based on our pharmacological and genetic data, we conclude that PI-dependent processes affect auxin-dependent vacuolar morphology.

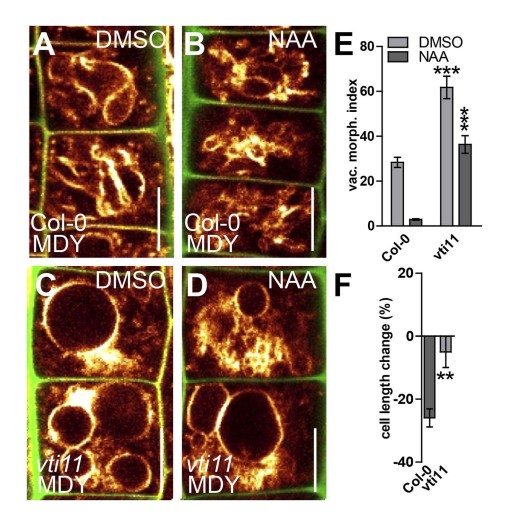

**Figure 6**. SNARE-dependent vacuolar morphogenesis is required for auxin regulated cell size determination. (**A–F**) Vacuolar morphology and cell size determination in the *vti11* mutant. Control treatment of *Col-0* with DMSO (**A**) or NAA (**B**) (250 nM; 20 hr). *vti11* mutants were treated with DMSO (**C**) or NAA (**D**) (250 nM; 20 hr). MDY-64 (orange) and propidium iodide (green) were used for confocal imaging of atrichoblast cells (**A–D**). (**E**) Vacuolar morphology (vac. morph. [μm²]) index of prior treatments in *Col-0* and *vti11* mutant. (**F**) Cell length change of *Col-0* and *vti11* atrichoblast cells after NAA (250 nM; 20 hr) treatment compared to DMSO control. n = 40 cells in eight individual seedlings for vacuolar morphology index quantification and n = 40 cells in 10 individual seedlings for cell size measurements. Error bars represent s.e.m. For statistical analysis either DMSO or NAA treatments were compared between control and indicated mutant/treated seedlings. Student's *t*-test p-values: **$p < 0.01$, ***$p < 0.001$. Scale bar: 15 μm.

The following figure supplements are available for figure 6:

**Figure supplement 1**. Auxin affects vacuolar morphology in several vacuolar *snare* single mutants.

**Figure supplement 2**. The vacuolar morphology in the *vti11* mutant remained differentially controlled in tricho-/atrichoblast root epidermal cells.

**Figure supplement 3**. The pVTI11:VTI11-GFP complements auxin phenotypes in *vti11* mutant.

We subsequently addressed whether the obstruction of PI homeostasis may impair auxin-dependent posttranslational regulation of SNARE proteins. Concomitant treatments with PI3/4-kinase inhibitor WM and auxin completely diminished the auxin effect on vacuolar SNAREs even at relatively high auxin levels (NAA [500 nM]) (*Figure 7G*; *Figure 7—figure supplement 1A–D*). This data suggests that WM sensitive processes interfere with both the auxin effect on vacuolar morphology and the auxin-dependent regulation of vacuolar SNARE abundance. Analogously, the *pi4kß1/2* double mutants showed reduced VAMP711-YFP abundance in untreated and auxin treated conditions (*Figure 7N*; *Figure 7—figure supplement 1E–H*). We accordingly conclude that auxin requires PI4K activity to modulate vacuolar SNAREs.

In contrast to WM, SNARE stability was still increased in response to high auxin (NAA [500 nM]) in *pi4kß1/2* double mutants (*Figure 7N*; *Figure 7—figure supplement 1E–H*). Notably, *pi4kß1/2* double mutants showed auxin sensitive vacuolar morphogenesis at these higher auxin levels (*Figure 7—figure supplement 1G,H*). Based on this data, we assume that PI4 kinase activity and possibly other, WM sensitive, PI-dependent processes affect auxin-dependent posttranslational regulation of vacuolar SNARE proteins. We conclude that the PI-dependent interference with the auxin effect on vacuolar SNAREs completely coincides with a reduced impact of auxin on vacuolar morphology and meristematic cell size.

## Auxin effect on vacuolar morphogenesis interrelates with auxin-dependent cellular growth inhibition

Here we show that auxin limits cellular vacuolisation, correlating with its negative impacts on meristematic cell size. Prolonged auxin treatments shift the cell length ratio of tricho- and atrichoblasts. Assuming that there is no cellular sliding, the auxin effect on meristematic cell size might not only depend on cellular expansion, but also on altered division rates. To exclude cell division and to assess whether the auxin effect on vacuolar morphology could limit cellular growth, we subsequently concentrated solely on cellular expansion of epidermal cells in the elongation zone and recorded their maximum expansion under high auxin conditions. As expected, auxin repressed cellular elongation in wild-type epidermal cells (*Figure 8A,B*). In contrast, pharmacological inhibition of PI3 and PI4 kinases led to reduced sensitivity to auxin-dependent inhibition of epidermal growth (*Figure 8C,D,I*). *pi4kß1/2* double mutants were similarly resistant to the auxin-dependent inhibition of cellular expansion (*Figure 8E,F,J*).

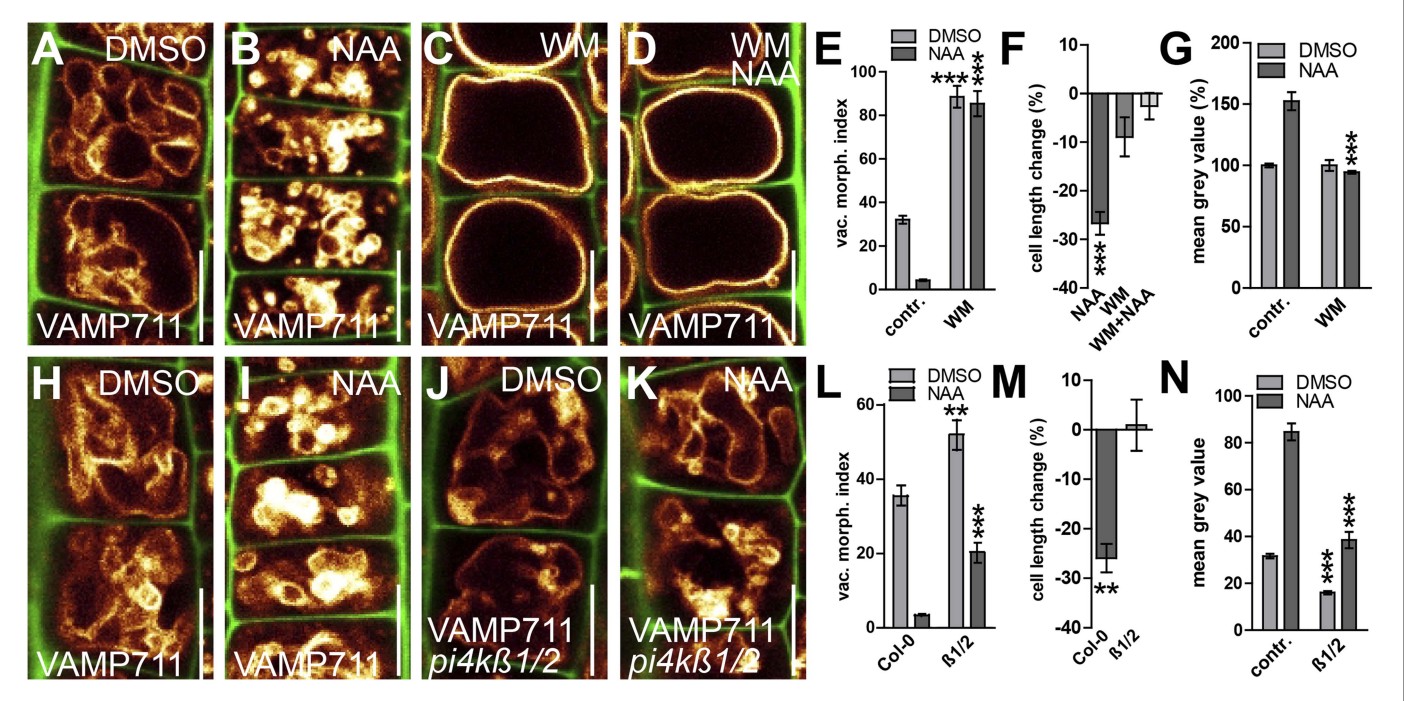

**Figure 7**. PI4-kinase function is required for auxin dependent vacuolar morphology, cell size determination and control of posttranslational VAMP711 abundance. (**A–G**) Effect of wortmannin (WM) on auxin regulated vacuolar morphology, cell growth inhibition and VAMP711 abundance. Control treatment of VAMP711-YFP atrichoblasts with DMSO (**A**) or NAA (**B**) (250 nM; 20 hr). VAMP711-YFP expressing seedlings after WM (**C**) (10 µM; 20 hr) or NAA/WM co-treatment (**D**). Quantification of vacuolar morphology (vac. morph. [µm²]) index (**E**) and cell length change (**F**). (**G**) Relative mean grey value of VAMP711-YFP abundance after NAA (500 nM; 20 hr) and/or WM (10 µM; 20 hr) treatment. Corresponding images are shown in *Figure 7—figure supplement 1*. (**H–M**) Effect on auxin regulated vacuolar morphology, cell growth inhibition and VAMP711 abundance in *pi4kß1/2* plants. Control treatment of VAMP711-YFP atrichoblasts with DMSO (**H**) or NAA (**I**) (100 nM; 20 hr) for comparability. VAMP711-YFP expression in *pi4kß1/2* mutant background after DMSO (**J**) or NAA (**K**) (100 nM; 20 hr) treatment. Quantification of vacuolar morphology (vac. morph. [µm²]) index (**L**) and cell length change (**M**). (**N**) Absolute mean grey value of VAMP711-YFP abundance after NAA (500 nM; 20 hr) treatment in the *pi4kß1/2* mutant background. Corresponding images are shown in *Figure 7—figure supplement 1*. VAMP711-YFP (orange) as a vacuolar marker and propidium iodide (green) for decorating the cell wall were used for confocal imaging of atrichoblast cells. n = 32 cells in eight individual seedlings for cell length measurements and n = 40 cells in eight individual seedlings for vacuolar morphology index quantification. Error bars represent s.e.m. For statistical analysis either DMSO or NAA treatments were compared between control and indicated mutant/treated seedlings. Student's *t*-test p-values: **$p < 0.01$, ***$p < 0.001$. Scale bar: 15 µm.

The following figure supplements are available for figure 7:

**Figure supplement 1**. PI4 kinases posttranslationally control VAMP711 abundance.

**Figure supplement 2**. Tricho-/atrichoblast cell length in wortmannin treated samples.

Subsequently, we tested root organ growth in response to auxin. Exogenous application of auxin strongly reduced the root length of wild-type seedlings, but was less effective following pharmacological (WM treated wild-type) and genetic interference (*pi4kß1/2* double) with PI kinases (*Figure 8L–N*). Notably and comparable with our cellular analysis, increased auxin concentrations led to root growth inhibition also in *pi4kß1/2* double mutants (*Figure 8—figure supplement 1A–D*). In conclusion, this data suggests that PI-dependent processes contribute to auxin-dependent inhibition of cellular and root organ growth.

Thereafter we likewise addressed auxin dependent inhibition of cellular expansion in VTI11 deficient roots. In agreement with our data on meristematic cell size control, cellular elongation was also less sensitive to auxin in *vit11* mutants (*Figure 8G,H,K*). The negative effect of auxin on root organ growth was also reduced in *vti11* mutants (*Figure 8L,O*), but was complemented by VTI11-GFP expression (*Figure 6—figure supplement 3*). Therefore, we conclude that auxin requires VTI11 function to inhibit cellular expansion and moreover root organ growth.

We have demonstrated that auxin interferes with SNARE abundance at the tonoplast, correlating with its effect on vacuolar morphology. What is striking is that auxin-dependent modulation of vacuolar shape precisely coincides with auxin dependent growth inhibition. Pharmacological and genetic interference with the auxin-dependent regulation of SNARE abundance corresponds with auxin resistant vacuolar morphology. Moreover, abrogation of the cellular auxin effect on vacuolar morphology interrelates with lower sensitivity to auxin-dependent inhibition of cellular growth.

We propose, based on these independent lines of evidence, that auxin signalling utilises SNARE-dependent vacuolar morphogenesis to restrict cellular expansion. Such regulation could have widespread developmental consequences, such as determining root organ growth rates.

## Discussion

It has been noted that the morphology of plant vacuoles correlate with cell size (*Owens and Poole, 1979*; *Berger et al., 1998*; *Löfke et al., 2013*) and, hence, it was tempting to postulate that vacuoles may even drive cellular growth (*Marty, 1999*). Surprisingly, little is actually known about mechanisms controlling vacuolar morphology and whether the vacuoles are indeed involved in growth regulation. Plant vacuoles, inter alia, are claimed to be important cellular osmoregulators, and accordingly, have been hypothesised as contributing to turgor-dependent cellular growth induction (*Marty, 1999*). This may explain the correlation between vacuolarisation with cellular size. Though appealing, also this hypothesis awaits experimental validation. Hence, the potential role of auxin in turgor regulation should also be carefully assessed. Just such a role has been insinuated in recent papers related to lateral root emergence. Auxin responses have been reported to reduce cellular pressure in cell files facing lateral root primordia (*Peret et al., 2012*) and vacuolar morphology alterations have been recently described in endodermal tissues during lateral root emergence (*Peret et al., 2012*; *Vermeer et al., 2014*). However, these proposed auxin responses are not related to cellular growth regulation and it remains

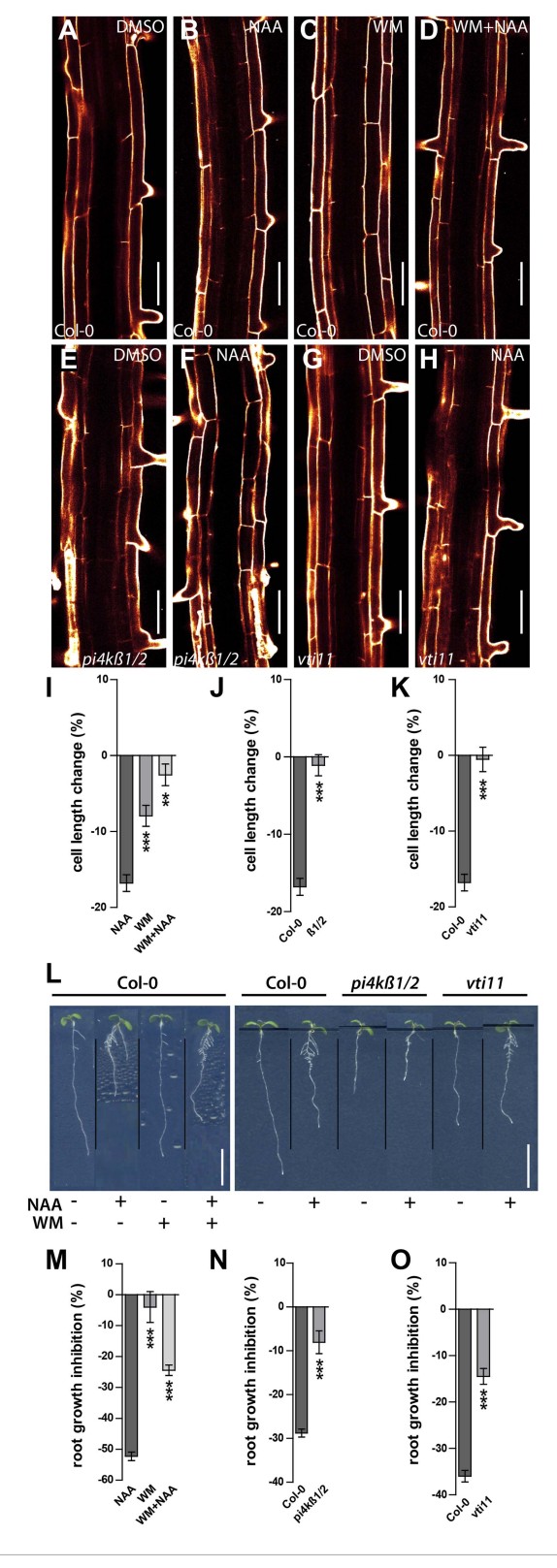

**Figure 8**. Auxin dependent vacuolar morphogenesis links auxin dependent growth inhibition. (**A–H**) Fully elongated root cells in the differentiation zone of *Col-0* after DMSO (**A**), NAA

*Figure 8. continued on next page*

*Figure 8. Continued*

(**B**) (250 nM), WM (**C**) (2 µM), WM+NAA (**D**) and *pi4kß1/2* after DMSO (**E**) or NAA (**F**) (250 nM), as well as *vti11* after DMSO (**G**) or NAA (**H**) (250 nM) treatments. (**I**) Cell length change of *Col-0* trichoblast cells after NAA (250 nM; 20 hr) and/or WM (2 µM; 20 hr) treatment. (**J**) Cell length change of *Col-0* and *pi4kß1/2* trichoblast cells after NAA (250 nM; 20 hr) treatment. (**K**) Cell length change of *Col-0* and *vti11* trichoblast cells after NAA (250 nM; 20 hr) treatment. (**L**) NAA mediated root growth inhibition of *Col-0*, *pi4kß1/2* and *vti11* after NAA (125 nM) and/or WM (2 µM) treatment. (**M**) Quantification of root growth inhibition in *Col-0* after NAA (125 nM) and/or WM (2 µM) treatment (germinated; 12 DAG). (**N**) Quantification of root growth inhibition *Col-0* and *pi4kß1/2* after NAA (125 nM) treatment (germinated; 8 DAG). (**O**) Quantification of root growth inhibition in *Col-0* and *vti11* after NAA (125 nM) treatment (germinated; 8 DAG). Propidium iodide (red) for decorating the cell wall was used for confocal imaging of epidermal cells. For cell length change n = 30 seedlings out of three independent experiments with approximately 90–120 quantified cells in total; and n = 20–25 roots for root growth inhibition. Error bars represent s.e.m. For statistical analysis either NAA or WM treatments were compared between control and/or indicated mutant/treated seedlings. Student's *t*-test p-values: *$p < 0.05$, ***$p < 0.001$. Scale bar: 50 µm (**A**–**H**); 1 cm (**L**).

The following figure supplement is available for figure 8:

**Figure supplement 1**. Root growth of *pi4kß1/2* mutants reacts to auxin in a dose dependent manner.

to be seen whether the auxin impact on vacuolar morphology described here is related to interference with turgor pressure.

Here we show that auxin controls the overall vacuolar morphology. Several pieces of independent evidence suggest that the modulation of the vacuolar SNAREs allows the phytohormone auxin to limit cellular growth. Future studies will further assess the precise molecular role of vacuolar SNAREs in shaping the plant vacuole. Auxin impacts on vacuolar SNAREs in a posttranslational manner and it is tempting to speculate that auxin could impose a conformational change on the SNARE complex, possibly affecting its stability and activity. Accordingly, the auxin effect on vacuolar SNAREs could impact on homotypic vacuolar fusion events, leading to smaller luminal structures. Alternatively, also a structural role of SNAREs has been envisioned (*Di Sansebastiano, 2013*), which could be yet another mechanism to impact on vacuolar organisation. Notably, a recent study suggested that most vacuolar structures are interconnected within an untreated cell (*Viotti et al., 2013*). Hence, it needs to be seen how precisely auxin impacts on vacuolar morphology and whether these structures are still interconnected in high auxin conditions.

We assume that auxin utilises the plant vacuole as a cellular effector to restrict cellular expansion. Hence, we conclude that previous correlative observations on cell size and vacuolar morphology (*Owens and Poole, 1979*; *Berger et al., 1998*; *Löfke et al., 2013*) may not reflect that vacuoles drive growth, but rather rely on a vacuole-dependent cellular mechanism that limits growth.

Our data extends the current view of auxin biology, suggesting that auxin coordinates extracellular and intracellular components, such as cell wall acidification (*Sauer and Kleine-Vehn, 2011*; *Spartz et al., 2014*) and vacuolar morphogenesis, for driving and restricting cellular growth. In this light, the luminal increase of plant vacuoles alone may not be sufficient to induce larger cell sizes (*Figure 6—figure supplement 2*; *Figure 7—figure supplement 2*) due to cell wall limitations. In contrast, limiting cellular vacuolarisation appears sufficient to restrict cellular growth. Such a dual growth mechanism would allow plants to dynamically de- and accelerate cellular expansion, integrating multiple distinct, possibly conflictive internal and external triggers.

## Materials and methods

### Plant material, growth conditions and DNA constructs

We used *Arabidopsis thaliana* of ecotype Columbia 0 (*Col-0*). *vamp711* (N673991), obtained from the Nottingham Arabidopsis stock centre and the decrease of *VAMP711* transcript was shown by RT-PCR. Other plant lines were described previously, *pUBQ10::VAMP711-YFP/RFP* (Wave 9Y/R) (*Geldner et al., 2009*), *pER8* and *pER8::YUC6* (*Mashiguchi et al., 2011*), *tir triple: tir1-1/afb1-3/afb3-4* (*Dharmasiri et al., 2005b*), *pi4kß1/2* (*Preuss et al., 2006*), *35S::SYP21-YFP* (*Robert et al., 2008*), *SYP22::SYP22-GFP* in *syp22* background (*Uemura et al., 2010*), *syp21* (*pep12*) (*Uemura et al., 2010*), *syp22* (*vam3-1*) (*Uemura et al., 2010*), *vti11* (zigzag) (*Yano et al., 2003*), *VTI11::GFP-VTI11* in *vti11* (*Niihama et al., 2005*), *NET4A::NET4A-GFP* (*Deeks et al., 2012*). The *pi4kß1/2* and *pER8::YUC6* were crossed into Wave 9Y, respectively. *tir triple* plants expressing Wave 9R were obtained by floral

dipping in *Agrobacterium tumefaciens* liquid cultured cells harbouring the Wave 9R vector and were subsequently selected on Basta (Bayer, Germany) supplemented MS plates. Seeds were stratified at 4°C for 2 days in the dark and grown on vertically orientated ½ Murashige and Skoog (MS) medium plates under a long-day regime (16 hr light/8 hr dark) at 20–22°C. Gateway cloning was used to construct pMDC7_B(pUBQ)::VAMP711 (Destination vector from *Barbez et al., 2012*). The *VAMP711* fragment was amplified with the high-fidelity polymerase 'I proof' (BioRad, CA, USA) from the Wave line vector harbouring VAMP711 (Wave 9Y). The primers were designed using life technologies OligoPerfect (www.lifetechnologies.com/oligoperfect) and are given below. The fragment obtained was introduced into the pDONR221 (Invitrogen, CA, USA). Coding *VAMP711* sequence from the entry vector was subsequently introduced in the gateway compatible destination vector pMDC7_B (pUBQ) using the Invitrogen LR clonase(+) and the resulting construct was transformed into *Col-0* plants by floral dipping in *Agrobacterium tumefaciens* liquid culture. The T1 generation obtained was selected on hygromycin B supplemented MS plates.

## Chemicals

All chemicals were dissolved in DMSO and were applied in solid ½ MS-medium. Only dyes were applied in liquid ½ MS-medium before imaging. 1-naphthaleneacetic acid (NAA) was obtained from Duchefa (Netherlands), 5-F-IAA, estradiol, FM4-64, L-kynurenine (Kyn) and propidium iodide (PI) from Sigma (MO, USA), wortmannin (WM) from Cayman Chemical (MI, USA), MDY-64 from life technologies (CA, USA) and auxinole was kindly provided by Ken-ichiro Hayashi (*Hayashi et al., 2012*).

## Phenotype analysis

For analysing the vacuolar morphology and cell length change, all the experiments were carried out on 7 days old seedlings. For analysing the vacuolar morphology index, optical confocal sections (above the cell nucleus) of the root epidermis were acquired and processed in imageJ. Images were taken in the late meristematic zone (position shown in *Figure 1—figure supplement 1*). The largest luminal structure in five tricho- and/or atrichoblast cells were quantified by measuring the longest to widest distance and processed by multiplying the values. Means and standard error were calculated and statistical significance was evaluated by the Student's *t*-test using graphpad (http://www.graphpad.com/quickcalcs/). Figures display a representative experiment (out of three independent repetitions) utilising eight individual roots. The cell length change in the late meristematic zone (position shown in *Figure 1—figure supplement 1*) was quantified in the median epidermal confocal section decorated with propidium iodide to visualise the cell wall. Cell length measurements in the late meristematic zone were performed in imageJ by quantifying the length of four tricho- and/or atrichoblast cells which were averaged. Means and standard error were calculated and statistical significance was evaluated by the Student's *t*-test using graphpad (http://www.graphpad.com/quickcalcs/). Figures show a representative experiment (out of three independent repetitions) utilising 10 individual roots. To estimate the position for the cell length measurements in the elongation zone, seedlings were stained in propidium iodide for 5 min and subsequently imaged choosing the position were no propidium iodide (0.02 mg/ml) entered the vasculature, showing fully developed endodermal diffusion barriers. Length measurements of epidermal root hair cells were performed in imageJ by quantifying the length of individual cells which were averaged. Means and standard error were calculated and statistical significance was evaluated by the Student's *t*-test using graphpad (http://www.graphpad.com/quickcalcs/). Figures show a representative experiment (out of three independent repetitions) utilising 10 individual roots. In each root approximately three cells were quantified. For analysis of the root length, seedlings grown in vertically orientated plates were scanned on a flat-bed scanner and measurements were performed in imageJ. Per condition, 20–25 seedlings were analysed (for each experiment) 8 days after germination. Means and standard error were calculated and statistical significance was evaluated by the Student's *t*-test using graphpad (http://www.graphpad.com/quickcalcs/). Figures show a representative experiment (out of three independent repetitions).

## Protein immunoblot analysis

Root samples (∼50 mg each) of 7 days old *pUBQ10::VAMP711-YFP* expressing seedlings were shock-frozen and homogenised in liquid nitrogen, then 150 µl extraction buffers (67 mM TRIS pH 6.8, 133 mM DTT, 2.7% SDS, 13% glycerol, 0.01% bromophenol blue) were added and immediately incubated at 95°C for 5 min. After centrifugation, 15 µl of the extracts were separated by SDS-PAGE

(10% gel) and blotted onto a 0.2 µm polyvinylidene difluoride membrane (BioRad). After blocking with a solution of 5% skim milk powder in TBS-T (150 mM NaCl, 10 mM TRIS/HCl pH = 8.0, 0.1% Tween 20) the membrane was probed with an anti-GFP antibody (JL-8; Clontech; Takara Bio Europe, Japan) diluted 1:2000 in skim milk TBS-T solution or anti-alpha-tubulin (B-5-1-1; Sigma) diluted 1:50000 in skim milk TBS-T solution. Horseradish peroxidase-conjugated goat anti-mouse antibody (Dianova, Germany) was employed as secondary antibody (1:20000). For detection, the SuperSignal West Pico chemiluminescent detection reagent (Thermo scientific, MA, USA) was used. Three independent root samples were quantified with imageJ and normalized to alpha-tubulin, means and standard error were calculated and statistical significance was evaluated by the Student's *t*-test using graphpad (http://www.graphpad.com/quickcalcs/). Figure shows a representative experiment (out of three independent repetitions).

## Microscopy

For live cell imaging, wherever possible, roots were mounted in a propidium iodide (PI) solution (0.02 mg/ml) for counterstaining the cell walls. MDY-64 and FM4-64 staining was performed as described (*Scheuring et al., 2015*). For 3D imaging, epidermal cells were recorded with a step size of 1 µm with approximately 17–20 single images. For image acquisition a Leica DM6000 CS, TCS AOBS confocal laser scanning microscope was used, equipped with a HCX PL APO CS 20.0 0.70 IMM UV or a HCX PL APO CS 63.0 × 1.20 WATER objective. Fluorescence signals of GFP (excitation 488 nm and emission 500 nm–546 nm), YFP (excitation 514 nm and emission 525 nm–578 nm), RFP (excitation 561 nm and emission 574 nm–618 nm), propidium iodide (excitation 561 nm and emission 577 nm–746 nm) and MDY-64 (excitation 458 nm and emission 465 nm–550 nm) were processed with the Leica software LAS AF 3.1 or with ImageJ (http://rsb.info.nih.gov/ij/) and data was statistically evaluated by Student's *t*-test using graphpad (http://www.graphpad.com/quickcalcs/). Representative images are shown.

## Primer list and destination vector

VAMP711_ATTB1_fwd, GGGGACAAGTTTGTACAAAAAAGCAGGCTCGATGGCGATTCTGTACGCC, rev, GGGGACCACTTTGTACAAGAAAGCTGGGTCTTAAATGCAAGATGGTAGAGTAGGTC; UBQ5 fwd, GACGCTTCATCTCGTCC, rev, GTAAACGTAGGTGAGTCCA.

Destination vector: pMDC7_B(pUBQ) (*Barbez et al., 2012*).

## Acknowledgements

We are grateful to N Geldner, H Kasahara, K Hayashi, A Nakano, E Nielson, N Raikhel, T Ueda, and PJ Hussey for providing published material; Niko Geldner, Christian Luschnig, Stephanie Robert, Michael Sauer, and Daniel Van Damme for critical reading of the manuscript and the BOKU-VIBT Imaging Centre for access and expertise. This work was supported by the Vienna Science and Technology Fund (WWTF), Austrian Science Fund (FWF) (Projects: P26568-B16 and P26591-B16) (to JK-V) and the Deutsche Forschungsgemeinschaft (DFG) (personal postdoctoral fellowships to CL and DS).

## Additional information

### Funding

| Funder | Grant reference | Author |
| --- | --- | --- |
| Vienna Science and Technology Fund | Vienna Research Group | Jürgen Kleine-Vehn |
| Deutsche Forschungsgemeinschaft | postdoctoral fellowships | Christian Löfke, David Scheuring |
| Austrian Science Fund | P26568-B16 | Jürgen Kleine-Vehn |
| Austrian Science Fund | P26591-B16 | Jürgen Kleine-Vehn |

The funders had no role in study design, data collection and interpretation, or the decision to submit the work for publication.

### Author contributions

CL, Conception and design, Acquisition of data, Analysis and interpretation of data, Drafting or revising the article; KD, DS, Acquisition of data, Analysis and interpretation of data, Drafting or

revising the article; JK-V, Conception and design, Analysis and interpretation of data, Drafting or revising the article

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
