## [Decision Letter]

Thank you for sending your work entitled “Auxin regulates SNARE-dependent vacuolar morphology restricting cell size” for consideration at *eLife*. Your article has been evaluated by Detlef Weigel (Senior editor) and 3 reviewers, as well as a member of our Board of Reviewing Editors.

The Reviewing editor and the reviewers discussed their comments before we reached this decision, and the Reviewing editor has assembled the following comments to help you prepare a revised submission.

As you can see from their comments, the reviewers have substantive concerns regarding your work, and we would like to ask you to address them comprehensively. Especially:

1) Please pay particular attention to the issues raised by reviewer 1, regarding the quantification of your markers. This issue is at the very heart of your manuscript, and we consider that it must be resolved to render your work publishable. Western blots for the other fusion proteins might be one way to provide more evidence, along with the reviewer's suggestions.

2) Please also clarify the involvement of TIR1-type auxin signaling, which suggests that there must be a transcriptional effect in this regulation somewhere. If it is not directly targeting the SNARE expression, but something regulating it, then do the time scales allow the conclusion that the auxin regulation is nevertheless direct and not a secondary effect?

3) The mutants that you present appear to have largely been published, nevertheless we would like to see evidence that the effects you report are due to those mutations, i.e. that you do not observe them in transgenically complemented plants.

*Reviewer #1*:

The authors describe an interesting series of experiments suggesting that auxin increases the abundance of the vacuolar SNARE protein VTI11 and leads to a contraction of the vacuole resulting in a complex multi-lobed structure. These two observations are linked by the fact that a non-functional mutant allele *vti11* does not support the formation of the compact multi-lobed structures. Overall, this paper could make an interesting contribution to *eLife*.

Critical points to be fixed:

The increase in VTI11 abundance in the analyzed cell types is not convincing. The authors show a strong increase in the signal of a YFP fusion protein upon NAA treatment and conclude that this means increased protein levels. This conclusion is challenged by the fact that artificial overexpression of VAMP711 did not influence vacuolar morphology and, more importantly, that the lipid marker MDY64 shows exactly the same NAA-induced increase in fluorescence as the VAMP11-YFP (Figure 1). This draws the validity of the microscopic assessment of SNARE abundance into question, particularly because the authors showed similar NAA-dependent increases in the abundance for a number of other vacuolar SNAREs, and even for *syp21*, which is actually on a pre-vacuolar compartment. My concern is that the contraction of the vacuoles into compact structures might lead to erroneous measures of the fusion proteins, due to the fact that the authors evaluated only a single confocal section, probably through the center plane of these cells, rather than a stack of images covering the entire depth. A consequence could be that in cells with voluminous vacuoles much of the vacuolar membranes lies outside the confocal section and will not be measured. In the contracted state, a greater share of this membrane should be closer to the center of the cell, leading to an increase in the fluorescence in the single confocal section.

The authors could rule out that their results are artefacts by addressing several points:

1) Integrate the fluorescence signals of the various SNARE proteins through a stack of sections covering the entire cell.

2) Do the same analysis for the lipid marker MDY64 and discuss the strong effect observed.

3) Present negative controls that do not show an NAA-induced increase of fluorescence. YFP-fusions of other vacuolar membrane proteins should be analyzed.

4) Re-decorate the blot in Figure 3 for several other vacuolar marker proteins in order to demonstrate specificity of the effect of VAMP11-YFP

*Minor comments*:

Many experiments use a YFP-fusion of VAMP11 and the fluorescence emission of YFP is very sensitive to pH changes. The authors should at least provide good arguments indicating that NAA does not change the intracellular pH to which the YFP tag is exposed.

Are the fluorescent protein tags on the cytosolic or on the lumenal side of the membrane?

It is not self-evident how an increase in vacuolar SNAREs could lead to a collapse to the vacuoles into a very compact morphology. The Discussion would be more stimulating if it presented at least some speculations on this issue.

The vacuolar morphology index appears too simplified in order to represent the complex morphological changes in a meaningful way. It arbitrarily selects only the largest vacuole in the picture as a measure, leaving aside the many small and medium-sized structures. Given this, the statistical analysis of such results appears as formally correct but of questionable relevance. I would have preferred to see an analysis of the total vacuolar volume in these cells, which may be the parameter that is most relevant for cell expansion.

*Reviewer #2*:

Löfke et al. use *Arabidopsis* root epidermis to show how auxin controls growth of growth-competent cells via stabilising protein components of the SNARE complex, which in turn regulate vacuole morphology. A range of genetic and chemical approaches has been taken and the results generally support the conclusions made by the authors. The manuscript is well written and the unravelling of this auxin-dependent control of vacuole morphology and cell growth will be interesting to a wide audience of cell and developmental biologists not restricted to the plant field. Below are some points for the authors to address, not in any particular order.

In the subsection headed “Vacuolar SNARE function is required for the auxin-dependent modulation of vacuolar morphology”, paragraphs 1 and 2 could be made clearer. In the reference to the redundant action of SNARE components, the *vti11* mutant is mentioned as displaying roundish vacuoles. Does this mean that *vti11* is the only mutant in a SNARE-encoding gene that has a phenotype? Also it would be useful to explain earlier what it means that VTI11 is a Qb-SNARE (see point on SNARE background below). Finally, since the vacuole morphology is clearly different in the *vti11* mutant from that in wild type, it is puzzling why the authors indicate that the defects in SNARE-dependent vacuolar morphology regulation in *vti11* are relatively mild. Perhaps they mean that the altered vacuolar morphology in *vti11* has little effect on cell growth in A and T cells?

The authors explain in the discussion that lack of increased growth by WM treatment (Figure 6—figure supplement 1) and in the atrichoblasts of the *vti11* mutant (Figure 5–figure supplement 2) could be due to cell wall limitations. I think they can expand their reasoning to include that the increased growth observed upon Kynurenin treatment in Figure 1 suggests that auxin acts not only on SNAREs/vacuolar morphology, but may also restrict growth by inhibiting availability of cell wall components.

Along the same line, it seems to me a discussion could be included on coordination between tricholast and atrichoblast cell files as well as cell divisions within a cell file. For example, do any of the genetic and pharmological treatments lead to a difference in the ratio of trichoblasts:atrichoblasts in the meristematic zone or total number of cells?

Figures 6 and 7 could be amalgamated into one figure. Figure 7 panels are identical to Figure 6 panels except for the higher amount of NAA. Same goes for Figure 7 and Figure 6. I suggest adding Figure 7 to Figure 6 and turn Figure 7 into Figure 6—figure supplement 2.

In the subsection headed “Auxin effect on vacuolar morphogenesis interrelates with auxin-dependent cellular growth inhibition” a paragraph describes the effect of VTI11 and PI4K activity on root epidermal cell length (elongation zone) and overall root growth. Two points: 1) It is mentioned in the legend to Figure 8 that trichoblasts were measured. Why not atrichoblasts? 2) It is surprising that WM treatment alone has a negative effect on growth (Figure 8). If anything, it would be expected to be positive. Can the authors explain this?

*Minor comments*:

Some of the headings could be made more specific. For example, “Interference with the auxin effect on vacuolar SNAREs impedes auxin-dependent cell size control”, this heading should refer to the connection made with PI-dependent processes.

I appreciate that space is limiting, but the minimal background information on the SNARE complex makes it difficult to relate to their role in this process. A few additional sentences to introduce SNARE would be useful. For example: 1) one gets the impression that SNAREs only exist in yeast and plants, although they indeed are common to all multicellular eukaryotic organisms, 2) mention their function in controlling membrane fusion, and 3) it would be useful to let the reader know what they are composed of such that the reader is prepared for what the Qb-SNARE VTI11 is.

*Reviewer #3*:

This paper reports on the dramatic effect auxin can have on the shape of root tip cell vacuoles. The data indicate that via auxin receptors, auxin can regulate the abundance of vacuolar SNARES in particular the QbSNARE VTI11. The authors have quantified the effect by designing a vacuolar morphology index, so that they can compare vacuolar morphology in the various mutants and treated root tips. The method of quantification is relatively simple and crude but does effectively reflect the changes in vacuolar morphology which in some cases are visually very striking. I'm sure with time the group will develop better methods of assaying the changed in vacuolar morphology.

In general the study is well presented and the paper well written and is a substantial contribution to the field. I feel that the authors have been a little cautious in their Discussion and have made little attempt to hypothesis as to why a vacuolar SNARE may have such a dramatic effect on vacuolar morphology. It is difficult to tell if in some of the micrographs we are looking at many individual vacuoles which have not fused together due to lack of SNARE complexes or whether they are all interconnected but just change in morphology. A bit of speculation would be good.

*Minor comments*:

In the beginning of the subsection headed “Vacuolar SNARE function is required for the auxin-dependent modulation of vacuolar morphology”, perhaps you need to introduce VTI11 here or at least put it in the section heading.

In the Discussion section, I really wonder whether turgor exerted by the vacuole in the root tip is important. The vacuole appears very pleomorphic at this stage of development.

---

## [Author Response]

*1) Please pay particular attention to the issues raised by reviewer 1, regarding the quantification of your markers. This issue is at the very heart of your manuscript, and we consider that it must be resolved to render your work publishable. Western blots for the other fusion proteins might be one way to provide more evidence, along with the reviewer's suggestions*.

We thank particularly reviewer 1 for raising these concerns. We accordingly paid particular attention to the quantification of vacuolar SNAREs and performed several experiments. 1) We experimentally excluded that membrane crowding accounts for the observed increase in fluorescent intensity. 2) Moreover, we performed detailed z-stack imaging, confirming our previous single section results on auxin induced increase in cellular abundance of vacuolar SNAREs. 3) In agreement, we also performed additional western blots, confirming that auxin increases the abundance of several vacuolar SNARE proteins. 4) Finally, we included an additional vacuolar marker NET4a-GFP in all our experiments, which fluorescent intensity or protein abundance is not affected by auxin. This set of experiments illustrates that auxin affects SNARE abundance at the tonoplast. The additional data is shown in Figure 4 and Figure 4—figure supplement 1.

*2) Please also clarify the involvement of TIR1-type auxin signaling, which suggests that there must be a transcriptional effect in this regulation somewhere*. *If it is not directly targeting the SNARE expression, but something regulating it, then do the time scales allow the conclusion that the auxin regulation is nevertheless direct and not a secondary effect?*

We once again thank you for the very valuable suggestion. We focused on the time scale of the auxin effect on vacuolar morphology as suggested. Accordingly, we emphasized on detailed time course experiments. First of all we reveal that the auxin effect on vacuolar morphology increases in time. Moreover, we could detect the first auxin induced morphological change of the vacuole within 15-30 minutes. This time frame is within in the time scale of fast genomic responses. Moreover, this finding also allowed us to illustrate that the auxin effect on vacuolar morphology preceded the auxin impact on cell size, which required ∼ 45 minutes. We included the data in the revised manuscript (Figure 2).

*3) The mutants that you present appear to have largely been published, nevertheless we would like to see evidence that the effects you report are due to those mutations, i.e. that you do not observe them in transgenically complemented plants*.

The interpretation of the *vti11* mutant phenotype is indeed central in our manuscript and, hence, we followed the suggestion, utilizing a functional pVTI11::VTI11-GFP construct to complement the *vti11* mutant phenotypes. Notably, VTI11-GFP expression in *vti11* mutant epidermal cells did not only rescue vacuolar morphology in untreated conditions, but also reversed auxin sensitivity in regards to vacuolar morphology and organ root growth. The new set of data and its quantification is now shown in Figure 6—figure supplement 3.

Reviewer #1:

*1) Integrate the fluorescence signals of the various SNARE proteins through a stack of sections covering the entire cell*.

We followed the reviewer’s suggestion and imaged again all used SNARE markers and integrated the signals through a stack of sections covering the entire cell. The new data confirms our previous single section data and is now shown in Figure 4—figure supplement 2. To further guide the reader on the exact optical sectioning in our figures (position for quantifications of 2D images), we also included a 3D overview screen of the root and VAMP711-YFP distribution in Figure 1—figure supplement 1.

*2) Do the same analysis for the lipid marker MDY64 and discuss the strong effect observed*.

We followed the reviewer’s suggestion and performed simultaneously image acquisition of VAMP711-Y/RFP and lipid markers MDY-64 and FM4-64. We did not detect any signal increase of MDY-64 or FM4-64 as compared to VAMP711-Y/RFP. The previous pictures must have been misleading and we apologize for not choosing representative pictures in the previous version. The new data is now shown in Figure 4—figure supplement 1.

*3) Present negative controls that do not show an NAA-induced increase of fluorescence. YFP-fusions of other vacuolar membrane proteins should be analyzed*.

We appreciate the suggestion and included the tonoplast marker NET4A-GFP, which was not affected by auxin. This suggests certain specificity for the auxin effect on vacuolar SNAREs. The new data is now shown in Figure 4.

*4) Re-decorate the blot in*
Figure 3
*for several other vacuolar marker proteins in order to demonstrate specificity of the effect of VAMP11-YFP*

We followed the reviewer’s suggestion and have performed western blot analysis including additional SNARE proteins and the NET4A-GFP (as a negative control). The data is now included in Figure 4.

Minor comments:

*Many experiments use a YFP-fusion of VAMP11 and the fluorescence emission of YFP is very sensitive to pH changes. The authors should at least provide good arguments indicating that NAA does not change the intracellular pH to which the YFP tag is exposed*.

We are grateful for this comment and assume many readers will have similar concerns. Therefore, we included a discussion on this aspect in the revised version of the manuscript. In our experiments we used VAMP711 fusion proteins to RFP and YFP. RFP and YFP proteins have very distinct sensitivities to pH changes, but both fusion proteins showed comparable auxin induced increase in fluorescent intensity. Moreover, previous work (11) suggested that exogenous application of auxin does not detectably affect cytosolic pH. Hence, we conclude that the observed effects are not likely related to changes in cytosolic pH.

*Are the fluorescent protein tags on the cytosolic or on the lumenal side of the membrane*?

The fluorescent tags are facing the cytosol. This information is now included in the revised version of the manuscript.

*It is not self-evident how an increase in vacuolar SNAREs could lead to a collapse to the vacuoles into a very compact morphology. The Discussion would be more stimulating if it presented at least some speculations on this issue*.

This is indeed a very interesting question. We assume that auxin could trigger a conformational change of SNAREs, which may affect its activity. Alternatively, vacuolar SNAREs could also provide a structural component required for vacuolar organisation. We currently investigate these options. At the current stage we can only speculate and we followed the reviewer’s suggestion and elaborated on our Discussion.

*The vacuolar morphology index appears too simplified in order to represent the complex morphological changes in a meaningful way. It arbitrarily selects only the largest vacuole in the picture as a measure, leaving aside the many small and medium-sized structures. Given this, the statistical analysis of such results appears as formally correct but of questionable relevance. I would have preferred to see an analysis of the total vacuolar volume in these cells, which may be the parameter that is most relevant for cell expansion*.

The vacuole is a highly complex structure, which is visually distinct in every cell. Long term auxin treatments and its effect on vacuolar shapes are very easily detectable by eye. However, shorter time points require a solid quantification method to capture morphological changes. We implemented the vacuolar morphology index to detect such major changes in vacuolar morphology. Notably, the measurement of the biggest vacuolar structure turned out to be a very reproducible approximation for vacuolar morphology. Importantly the measured values are quite stable in between independent experiments (both untreated and treated). Moreover, the index is quite sensitive to auxin induced morphological changes, which guided us to depict earliest morphological changes following short-term auxin treatments. Notably, the vacuolar morphology index robustly correlates with the auxin effect on cell sizes. Therefore, we feel that this simple quantification method is straightforward and very suitable for the here conducted work. In this respect, we kindly agree also with reviewer 3 (“The method of quantification is relatively simple and crude but does effectively reflect the changes in vacuolar morphology which in some cases are visually very striking.”). Said that we also would like to mention that we would indeed further develop tools to depict and quantify the plant vacuoles. In our on-going research, we are also investigating the potential effect of auxin on vacuolar volume and whether it is a parameter that is most relevant. We will develop in the future more sophisticated 3D imaging approaches, which will allow us to address additional questions, such as raised by reviewer 1.

Reviewer #2:

*In the subsection headed “Vacuolar SNARE function is required for the auxin-dependent modulation of vacuolar morphology”,paragraphs 1 and 2 could be made clearer. In the reference to the redundant action of SNARE components, the* vti11 *mutant is mentioned as displaying roundish vacuoles. Does this mean that* vti11 *is the only mutant in a SNARE-encoding gene that has a phenotype? Also it would be useful to explain earlier what it means that VTI11 is a Qb-SNARE (see point on SNARE background below).*

We thank the reviewer for pointing out the lack of discussion and improved the Introduction and Discussion on SNARE proteins. It has been previously shown that several SNARE complex components indeed function redundantly. Some mutants display defects in storage vacuoles and mild growth phenotypes (e.g. Shirakawa et al., 2010). However, based on our analysis most mutants did not show any defects in vacuolar morphology.

*Finally, since the vacuole morphology is clearly different in the* vti11 *mutant from that in wild type, it is puzzling why the authors indicate that the defects in SNARE-dependent vacuolar morphology regulation in* vti11 *are relatively mild. Perhaps they mean that the altered vacuolar morphology in* vti11 *has little effect on cell growth in A and T cells?*

This formulation was indeed confusing. We intended to point out that in the *vti11* mutant vacuolar morphology is still differentially regulated in tricho- and atrichoblast. In the revised version of the manuscript, we reformulated the sentence: “Despite these apparent defects, vacuoles remained differentially controlled in *vti11* mutant tricho- and atrichoblast cells, indicating that the cell type-dependent regulation of vacuolar morphology is at least partially operational in *vti11* mutants.”

*The authors explain in the discussion that lack of increased growth by WM treatment (*Figure 6—figure supplement 1*) and in the atrichoblasts of the* vti11 *mutant (Figure 5–figure supplement 2) could be due to cell wall limitations. I think they can expand their reasoning to include that the increased growth observed upon Kynurenin treatment in*
Figure 1
*suggests that auxin acts not only on SNAREs/vacuolar morphology, but may also restrict growth by inhibiting availability of cell wall components.*

Yes, we agree with the reviewer, this data is in agreement with a dual role of auxin. Upcoming research in our group will further decipher this interplay.

*Along the same line, it seems to me a discussion could be included on coordination between tricholast and atrichoblast cell files as well as cell divisions within a cell file. For example, do any of the genetic and pharmological treatments lead to a difference in the ratio of trichoblasts:atrichoblasts in the meristematic zone or total number of cells*?

Indeed pharmacological treatments (high and low auxin conditions) lead to a change in the ratio of tricho- and atrichoblasts (Figure 9). We currently assume that plant tissues do not show cellular sliding due to the cell wall. Hence, cell division could contribute to the observed differences (especially after prolonged auxin treatments). This was also the reasoning to include analysis that assesses auxin-dependent growth repression in the elongation zone. We improved the underlying discussion in the revised version of the manuscript. In fact, we are currently testing whether we could use light sheet-based microscopy to follow prolonged auxin treatments in time. This approach would allow us to perform the required life cell imaging and to reveal or exclude the vacuolar requirements for cellular division.

Author response image 1.**DOI:**
http://dx.doi.org/10.7554/eLife.05868.025

Figures 6 and 7
*could be amalgamated into one figure.*
Figure 7
*panels are identical to*
Figure 6
*panels except for the higher amount of NAA. Same goes for*
Figure 7
*and*
Figure 6*. I suggest adding*
Figure 7
*to*
Figure 6
*and turn*
Figure 7
*into*
Figure 6—figure supplement 2.

We followed the advice and amalgamated Figures 6 and 7 into one figure and created a new figure supplement (Figure 6—figure supplement 3).

*In the subsection headed “Auxin effect on vacuolar morphogenesis interrelates with auxin-dependent cellular growth inhibition” a paragraph describes the effect of VTI11 and PI4K activity on root epidermal cell length (elongation zone) and overall root growth. Two points: 1) It is mentioned in the legend to*
Figure 8
*that trichoblasts were measured. Why not atrichoblasts*?

These were rather technical reasons. Visual limitations of single sections imposed the practical depiction of trichoblast cells. We used medium sections to depict the start of the differentiation zone (functional endodermal diffusion barrier). In these sections, a cell “without” hairs could be still a trichoblast cell whose root hair was simply not in focus. Therefore, we considered for our measurments only cells showing a root hair, which ensured that we measured only one cell type. We assume that relative tissue expansion is constant between tricho- and atrichoblasts. Therefore, the relative outcome would be the same if we had used atrichoblasts.

*And 2) It is surprising that WM treatment alone has a negative effect on growth (*Figure 8*). If anything, it would be expected to be positive. Can the authors explain this*?

Besides its effect on vacuolar morphology, WM interferes with several intracellular processes and it is likely to interfere with growth. It indeed has been previously shown that high levels of WM (33µM) have a negative impact on growth (Jaillais et al., 2006). We used very low WM concentrations and observed only a mild effect on growth. Besides its mild effect on growth, this concentration range did not only reveal auxin resistant vacuoles, but also reduced the auxin effect on cell size and root organ growth.

Minor comments:

*Some of the headings could be made more specific. For example, “Interference with the auxin effect on vacuolar SNAREs impedes auxin-dependent cell size control”, this heading should refer to the connection made with PI-dependent processes*.

We thank the reviewer for this suggestion and revised the heading accordingly: “Interference with phosphatidylinositol homeostasis affects vacuolar SNAREs and impedes auxin-dependent cell size control.”

*I appreciate that space is limiting, but the minimal background information on the SNARE complex makes it difficult to relate to their role in this process. A few additional sentences to introduce SNARE would be useful. For example: 1) one gets the impression that SNAREs only exist in yeast and plants, although they indeed are common to all multicellular eukaryotic organisms, 2) mention their function in controlling membrane fusion, and 3) it would be useful to let the reader know what they are composed of such that the reader is prepared for what the Qb-SNARE VTI11 is*.

We appreciate the very valuable suggestion and included more detailed background information on the SNAREs in the revised version of the manuscript.

Reviewer #3:

*I feel that the authors have been a little cautious in their Discussion and have made little attempt to hypothesis as to why a vacuolar SNARE may have such a dramatic effect on vacuolar morphology*.

We thank the reviewer for this suggestion. We indeed were very cautious in the previous version, because we did not directly address these questions. However, we agree with the reviewer and substantiated the discussion on vacuolar SNAREs in the revised version of the manuscript.

*It is difficult to tell if in some of the micrographs we are looking at many individual vacuoles which have not fused together due to lack of SNARE complexes or whether they are all interconnected but just change in morphology. A bit of speculation would be good*.

We again thank the reviewer to address this very interesting question. We included a paragraph discussing the possibility of interconnected vacuoles and cited also the latest results by Viotti et al., 2013. Ongoing research in our group is actually addressing this eminent question.

Minor comments:

*In the beginning of the subsection headed “Vacuolar SNARE function is required for the auxin-dependent modulation of vacuolar morphology”, perhaps you need to introduce VTI11 here or at least put it in the section heading*.

We followed the reviewer’s suggestion and included VTI11 in the section heading.

*In the Discussion section, I really wonder whether turgor exerted by the vacuole in the root tip is important. The vacuole appears very pleomorphic at this stage of development*.

We agree with the reviewer and discussed this issue accordingly: “Hence, the potential role of auxin in turgor regulation should also be carefully assessed. Just such a role has been insinuated in recent papers related to lateral root emergence. Auxin responses have been reported to reduce cellular pressure in cell files facing lateral root primordia (24) and vacuolar morphology alterations have been recently described in endodermal tissues during lateral root emergence (35, 24). However, these proposed auxin responses are not related to cellular growth regulation and it remains to be seen whether the auxin impact on vacuolar morphology described here is related to interference with turgor pressure.”